# SPADE : TRAINING-FREE IMPROVEMENT OF SPATIAL FIDELITY IN TEXT-TO-IMAGE GENERATION

## ABSTRACT

Text-to-Image (T2I) generation models have seen progressive improvements in their abilities to generate photo-realistic images. However, it has been demonstrated that they struggle to follow reasoning-intensive textual instructions, particularly when it comes to generating accurate spatial relationships between objects. In this work, we present an approach to improve upon the above shortcomings of these models by leveraging spatially accurate images (LSAI) as grounding reference to guide diffusion-based T2I models. Given an input prompt containing a spatial phrase, our method involves symbolically creating a corresponding synthetic image, which accurately represents the spatial relationship articulated in the prompt. Next, we use the created image alongside the text prompt, in a training-free manner to condition image synthesis models in generating spatially coherent images. To facilitate our LSAI method, we create SPADE, a large database[1] of 190k text-image pairs, where each image is deterministically generated through open-source 3D rendering tools encompassing a diverse set of 80 MS-COCO objects. Variation of the images in SPADE is introduced through object and background manipulation as well as GPT-4 guided layout arrangement. We evaluate our method of utilizing SPADE as T2I guidance on Stable Diffusion and ControlNet, and find our LSAI method substantially improves upon existing methods on the VISOR benchmark. Through extensive ablations and analysis, we analyze LSAI with respect to multiple facets of SPADE and also perform human studies to demonstrate the effectiveness of our method on prompts which contain multiple relationships and out-of-distribution objects. Finally, we present our SPADE Generator as an extendable framework to the research community, emphasizing its potential for expansion.

## 1 INTRODUCTION

The emergence of generative models in computer vision and natural language processing (Brown et al., 2020; OpenAI, 2023) have opened up a plethora of real-world applications. Text-to-image (T2I) models such as Stable Diffusion (Rombach et al., 2021) and DALL-E 2 (Ramesh et al., 2022) are one such class of models that generate images given an input text prompt. These models have attracted significant attention because of their capability to generate intricate and highly realistic images in response to complex textual prompts. As a result, they have been leveraged for complementary tasks such as image editing (Hertz et al., 2022) and image-to-image translation (Parmar et al., 2023).

Despite the plaudits, many studies have shown that these T2I methods fall short in their ability to precisely follow textual instructions, and fail to maintain compositionality (Feng et al., 2023a; Wang et al., 2023). In particular, VISOR (Gokhale et al., 2023) benchmarks a commonplace issue found in these models, which is their inability to consistently generate images that accurately reflect the spatial relationships mentioned in the input prompts. As shown in Figure 1, existing T2I models face two significant challenges in this context: a) they frequently struggle to generate all the objects mentioned in the text, and b) they often produce images with incorrect spatial arrangements. These failures can be attributed to the models' inability to generalize to object pairs and arrangements that

---

[1]We refer to it as a database and not as a dataset as it is not used for learning in this work.

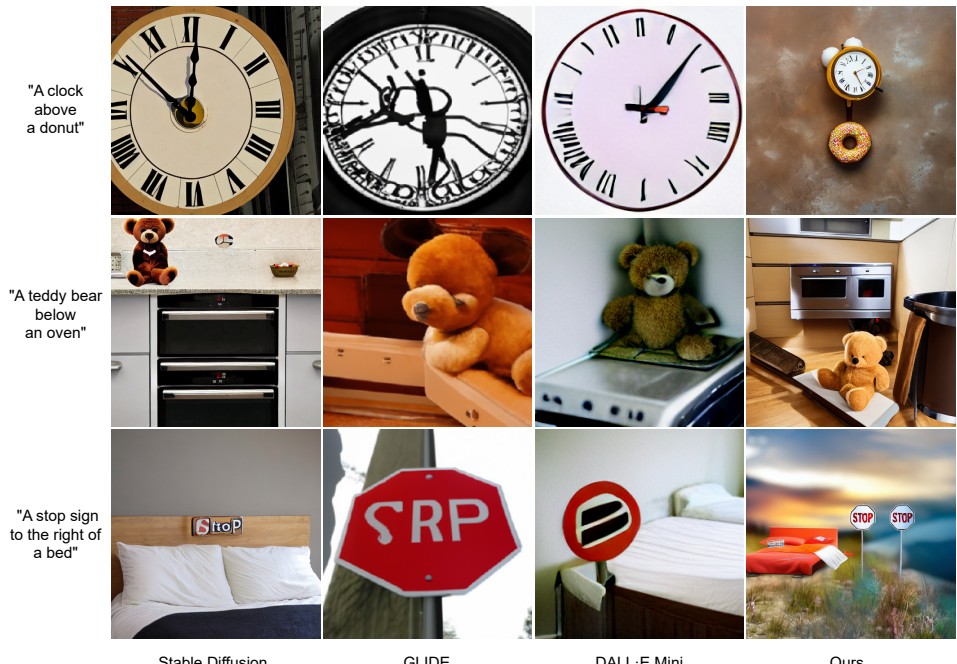

Figure 1: Traditional T2I models struggle to generate correct spatial relationships mentioned in the input prompt, often unable to generate all objects in the prompt. We present a training-free, guidance-based mechanism to address this shortcoming, outperforming existing methods on the VISOR benchmark.

are not encountered during training. It is even more difficult for these models to generate images of rare situations, e.g. *"a stop sign to the right of a bed"*.

A number of approaches, including Control-GPT (Zhang et al., 2023b) and Layout Guidance (Chen et al., 2023) have been proposed to address this issue. However, these approaches require either expensive training, or labeled annotations. Cognizant of these issues, we propose a simple yet effective approach, LSAI (**L**everaging **S**patially **A**ccurate **I**mages), that specifically aims at improving state-of-the art T2I models in their ability to generate spatially faithful images.

Our method first generates a symbolic reference image given an input prompt and then uses that reference image as additional guidance in a training-free paradigm. Specfically, we parse the input prompt in order to deterministically synthesize an image that exactly matches the input in terms of both objects and their spatial arrangement. These reference images are created via the SPADE (**SPA**tial Fi**DE**lity) Generator, an extendable framework which comprises Blender, a 3D rendering engine, complemented by additional modules such as the scene synthesizer and position diversifier to enhance generalization. Using the generator, we introduce SPADE, a large-scale database of 190k text-image pairs, where each image is spatially accurate to its corresponding text prompt.

Utilizing SPADE as guidance for state-of-the-art models such as Stable Diffusion (Rombach et al., 2021) and ControlNet (Zhang et al., 2023a), we find that our approach outperforms existing methods on the VISOR benchmark. Our VISOR Conditional and Unconditional scores are 97.72% and 53.08% respectively, with an Object Accuracy of 54.33%. More interestingly, we find that for a given text prompt, we are consistently able to produce spatially accurate images; a criterion that the majority of existing approaches fall short of. Finally, we underline the generalization capabilities of our approach by evaluating it on out-of-distribution and multi-object settings. To summarize, our contributions are as follows:

- We propose SPADE, a multi-faceted database of 190k text-image pairs, where each image is guaranteed to follow the spatial instruction mentioned in the text. The SPADE Generator is an extendable framework which currently covers 80 MS-COCO objects, 3 diverse backgrounds, and includes GPT-4 as an additional coordinate generator.

- We propose LSAI, a training-free T2I method that leverages SPADE as additional guidance for T2I models. We demonstrate state-of-the art performance in generating spatially faithful images, outperforming existing open-source methods on the VISOR benchmark.

- We examine the trade-off between diversity and controllability introduced by SPADE on T2I generation. Through human studies, we discover that our method is able to generalize to objects not included in SPADE as well as to prompts that contain multiple spatial directions.

## 2 RELATED WORKS

**Generative Models for Image Synthesis** The high dimensional and complex nature of images has led to image synthesis being viewed through many lenses over the years. Generative Adversarial Network (GAN) (Goodfellow et al., 2020; Gulrajani et al., 2017; Metz et al., 2017) based methods produce images of high quality, but suffer from optimization constraints and are unable to capture the complete data distribution. Auto-regressive models (ARM) (Chen et al., 2020a) and Variational Auto-Encoders (VAE) (Sohn et al., 2015) systems suffer from computationally demanding architectures and sampling quality issues, respectively. AlignDRAW (Mansimov et al., 2016) was the pioneering work that attempted to generate images from natural language captions. GLIDE (Nichol et al., 2022) adopts classifier-free guidance in T2I and explores the efficacy of CLIP Radford et al. (2021) as a text encoder. Compared to GLIDE, Imagen (Saharia et al., 2022) adopts a frozen language model as the text encoder, reducing computational overhead, allowing for usage of large text-only corpus. The emergence of Stable Diffusion and DALL-E has ignited substantial public curiosity in T2I generation. Therefore, it is imperative to ensure that these models become more robust and enhance their capacity for advanced reasoning.

**Synthetic Images for Vision & Language** The flexibility and control provided during creation of synthetic images have led to researchers exploring them for various visuo-linguistic benchmarks. CLEVR (Johnson et al., 2017) pioneered the utilization of synthetic objects in simulated scenes for visual compositionality reasoning. Many variants of CLEVR such as CLEVR-Hans (Stammer et al., 2021), CLEVR-Hyp (Sampat et al., 2021) and CLEVRER (Yi et al., 2019) probe multiple facets of visuo-language understanding with synthetic images and videos. PaintSkills introduced in DALL-EVAL (Cho et al., 2022) is a evaluation dataset that measures multiple aspects of a T2I model, which includes Spatial Reasoning, Image-Text Alignment and Social Biases. Compared to PaintSkills, SPADE is primarily leveraged as additional conditioning for better spatial alignment. Furthermore, we design SPADE to cover all 80 MS-COCO objects in a diverse manner across randomly generated backgrounds.

**Controllable Image Generation** To achieve better control over diffusion-based T2I methods, multiple methods have been proposed. ReCo (Yang et al., 2023) introduces learnable position tokens as part of its input allowing for precise region level control in the image. SpaText (Avrahami et al., 2023) takes annotated segmentation maps as additional inputs and learns a CLIP image embedding based spatio-temporal representation for accurate image generation. GLIGEN (Li et al., 2023) performs open-world grounded image generation by injecting captions and bounding boxes as additional grounding information. LayoutGPT (Feng et al., 2023b) uses LLMs to create layouts in the form of CSS structures, and then uses layout-to-image models to create both 2D and 3D indoor scenes. Layout Guidance (Chen et al., 2023) provides a test time adaptation by restricting specific objects to their bounding box location through modification of cross-attention maps. However, a shortcoming of this approach is the need of annotated bounding box locations which might not always be available. Control-GPT (Zhang et al., 2023b) first prompts GPT-4 to generate TikZ code which generates a sketch representation, given an input prompt. Followed by this, a ControlNet model is finetuned with the sketches, input prompt and grounding tokens to generate images. Although Control-GPT performs well on the VISOR benchmark, their method is expensive to train and is not always guaranteed to produce the correct TikZ code.

## 3 THE SPADE DATABASE

We introduce SPADE, a large database of text-image pairs, designed for better spatial relation understanding of T2I models. SPADE features a diverse collection of synthesized images as references

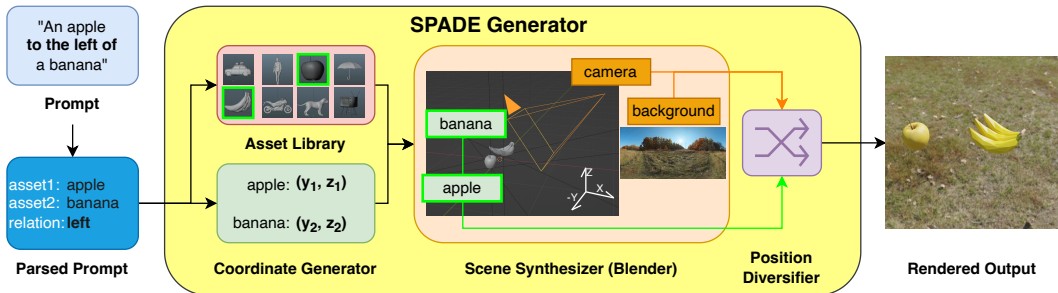

Figure 2: Given a text prompt, our SPADE Generator parses the objects and spatial relationship from it. Next, it deterministically synthesizes a reference image in the 3D scene, by placing the identified object assets at locations according to the parsed spatial relationship.

for image generation during the T2I process. We also offer the SPADE Generator that creates the reference images in the SPADE database, by extracting all relevant objects and placing them faithfully according to the spatial relation embedded in the given input prompt. In this paper, we study texts of precisely 2 objects and 1 spatial relation, where the 2 objects are placed either in a horizontal or vertical manner.

Figure 2 illustrates the SPADE Generator pipeline. In general, a reference image is a captured camera view of a synthesized scene in 3D, where 2 object models are positioned according to the spatial relation in the input prompt. The SPADE Generator consists of the following modules to curate and synthesize reference images for the SPADE database.

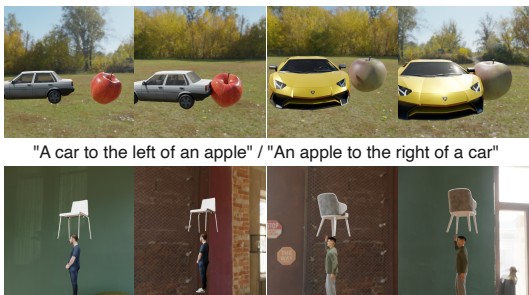

"A car to the left of an apple" / "An apple to the right of a car"

"A chair above a person" / "A person below a chair"

Figure 3: Altered object postures generated by the Position Diversifier module in the SPADE Generator. Even when the object assets and the background are identical, we still end up with visibly different reference images.

**Asset Library** The Asset Library includes a pre-set collection of 3D model assets depicting a wide range of realistic objects, with each object having multiple asset variants in textures and postures. Given an object name extracted from the input prompt, the Asset Library randomly selects one matching asset to be synthesized into the output. All asset models are rescaled to a universal height to ensure they are sufficiently visible in the final output.

**Coordinate Generator** Given the objects and the spatial relation from the prompt, the Coordinate Generator creates sets of corresponding numerical coordinates along the horizontal Y-axis and the vertical Z-axis to place the objects into a 3D environment. To make sure that the majority of the two objects can be captured, the coordinate values on the Y and Z axes are within the range of $[-100, 100]$. We randomly place one of the objects in the given range, and constrain the positioning of the other object based on the spatial relationship. The horizontal and vertical relationships are mapped to the Y and Z-axis respectively, while the objects' coordinates on X-axis are fixed at 0.

Alternatively, we also experiment with generating the objects' coordinates using GPT-4. We first feed GPT-4 a designed context prompt that includes specific example coordinates for each possible spatial relation. We then feed in an input prompt of 2 objects and 1 spatial relation in order to obtain the two sets of coordinates for placing the mentioned objects. Example prompts and responses are provided in Appendix C.

**Scene Synthesizer** The Scene Synthesizer builds a 3D scene of 4 major components: 2 objects, a background, and a camera. We place the object assets retrieved from the Asset Library at their respective coordinates obtained by the Coordinate Generator. We also set up the background using a 360-degree panorama image, which is a large sphere with interior textures centered at the 0-point

of origin. The camera component renders the scene from its view into SPADE. We place the camera along the positive X-axis at a specific distance from the two objects and aim its view at the 0-point.

**Position Diversifier** We lastly incorporate generalizability before rendering a synthesized scene. Figure 3 demonstrates how we introduce diversified appearances to all components in a scene. To change the orientations of the object assets, we add random small rotations along the Z-axis. We slightly alter the distance in between the objects so that they are not always symmetric around the 0-origin point. The background panorama image is freely rotated along the Z-axis, giving us an infinite number of static background options. In order to further diversify the perspective sizes and tilts of the object assets within the camera's view, we also add minor random nudges to the position and orientation of the camera.

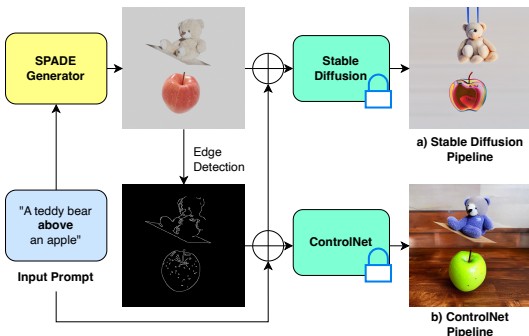

Figure 4: Our training-free method LSAI, takes an user-provided input prompt ($T$) and the corresponding reference image $x^{(g)}$ as input. With both the inputs, we perform diffusion-based image synthesis to generate spatially accurate images.

**Database Statistics** We incorporate 375 3D model assets across 80 MS-COCO (Lin et al., 2014) objects, with each object being linked to 3 to 5 royalty-free assets sourced from Sketchfab [2]. For an ordered object pair $(A, B)$, we consider two types of 2D relations, horizontal and vertical. Prompt sentences are generated in a templatized manner similar to Gokhale et al. (2023). We utilize 3 background panoramas [Indoor, Outdoor, White] from Poly Haven [3] and generate 5 text-image variants for every possible object pair, In total, we yield $^{80}P_2 \times 2 \times 5 \times 3 = 189,600$ text-image pairs.

## 4 METHOD

We introduce LSAI (**L**everaging **S**patially **A**ccurate **I**mages) for T2I generation. Our method takes as input a user-provided input prompt ($T$) and the corresponding reference image ($x^{(g)}$), both of which are generated via SPADE. Followed by this, we perform image synthesis to generate ($I$), i.e. $\phi(I|x^{(g)}, T)$, where $\phi$ is the image synthesis module. In our method, we make use of two independent training-free pipelines based on Stable Diffusion and ControlNet. We illustrate our method in Figure 4.

$$x(t) = x(t + \Delta t) + (\sigma^2(t) - \sigma^2(t + \Delta t))s_\theta(x(t), t)) + \sqrt{(\sigma^2(t) - \sigma^2(t + \Delta t))}\mathbf{z}, \quad (1)$$

Standard de-noising diffusion methods such as Stable Diffusion solve the reverse Stochastic differential equation (SDE) Anderson (1982); Song et al. (2020) 1 to approximate $x(0)$ by gradually de-noising $x(t)$, where, $\sigma(t)$ is a function that denotes the magnitude of the noise z ($z \sim \mathcal{N}(0, \mathbf{I})$) and $s_\theta(x(t), t)$ is the parameterized score function. SDEdit (Meng et al., 2022) approximated the reverse SDE process from $t_0 \in (0, 1)$, contrary to other methods that start from $t = 1$. Specifically, SDEdit starts from a guide image ($x^{(g)}$, in our case), selects $t_0$, then adds Gaussian noise of standard deviation $\sigma^2(t_0)$ and finally solves 1 to produce the synthesized $x(0)$. We leverage SDEdit into our Stable Diffusion pipeline, and perform image generation guided by $x^{(g)}$. We measure the influence of $t_0$ in our experiments and assess the balance it strikes between achieving photo-realism and maintaining spatial faithfulness.

ControlNet allows for fine-grained control over Stable Diffusion via low-level semantics such as edges, depth and segmentation maps, by making a trainable copy of the network which learns the additional conditioning. We leverage this backbone to demonstrate two key points: firstly, our reference images provide enough spatial information even when extracting low-level features from them, and secondly, we can mitigate any attribute-related biases present in the assets.

---

[2]https://sketchfab.com
[3]https://polyhaven.com/hdris

| Method | OA (%) | VISOR (%) | | | | | |
|---|---|---|---|---|---|---|---|
| | | uncond | cond | 1 | 2 | 3 | 4 |
| GLIDE | 3.36 | 1.98 | 59.06 | 6.72 | 1.02 | 0.17 | 0.03 |
| DALLE-mini | 27.10 | 16.17 | 59.67 | 38.31 | 17.50 | 6.89 | 1.96 |
| DALLE-v2 | **63.93** | 37.89 | 59.27 | 73.59 | 47.23 | 23.26 | 7.49 |
| SD 1.4 | 29.86 | 18.81 | 62.98 | 46.60 | 20.11 | 6.89 | 1.63 |
| Layout Guidance | 40.01 | 38.80 | 95.95 | - | - | - | - |
| Control-GPT | 48.33 | 44.17 | 65.97 | 69.80 | 51.20 | 35.67 | 20.48 |
| SD 1.4 + SPADE | 53.96 | 52.71 | 97.69 | 77.79 | 61.02 | 44.90 | 27.15 |
| SD 1.5 + SPADE | 54.33 | 53.08 | **97.72** | 78.07 | 61.27 | 45.44 | 27.55 |
| SD 2.1 + SPADE | 48.26 | 47.11 | 97.61 | 76.07 | 55.75 | 37.10 | 19.53 |
| ControlNet + SPADE | 56.88 | **55.48** | 97.54 | **78.82** | **62.93** | **48.58** | **31.59** |

Table 1: **Results on the VISOR Benchmark**. Leveraging SPADE as additional guidance for T2I models, we are able to achieve state-of-the art performance on the VISOR Benchmark.

| Method | VISOR_cond (%) | | | | | Object Accuracy (%) | | | | |
|---|---|---|---|---|---|---|---|---|---|---|
| | left | right | above | below | $\sigma_{Vc}$ | left | right | above | below | $\sigma_{OA}$ |
| GLIDE | 57.78 | 61.71 | 60.32 | 56.24 | 2.46 | 3.10 | 3.46 | 3.49 | 3.39 | 0.18 |
| DALLE-mini | 57.89 | 60.16 | 63.75 | 56.14 | 3.29 | 22.29 | 21.74 | 33.62 | 30.74 | 5.99 |
| DALLE-v2 | 56.47 | 56.51 | 60.99 | 63.24 | 3.38 | **64.30** | **64.32** | **65.66** | **61.45** | 1.77 |
| SD 1.4 | 64.44 | 62.73 | 61.96 | 62.94 | 1.04 | 29.00 | 29.89 | 32.77 | 27.80 | 2.12 |
| Control-GPT | 72.50 | 70.28 | 67.85 | 65.70 | 2.95 | 49.80 | 48.27 | 47.97 | 46.95 | 1.18 |
| SD 1.4 + SPADE | 97.53 | 97.45 | **98.09** | 97.66 | 0.29 | 52.42 | 52.11 | 56.93 | 54.38 | 2.22 |
| SD 1.5 + SPADE | 97.57 | **97.53** | 98.05 | 97.70 | 0.24 | 52.99 | 52.59 | 56.80 | 54.92 | 1.94 |
| SD 2.1 + SPADE | **97.81** | 97.46 | 97.91 | 97.28 | 0.30 | 46.70 | 47.94 | 49.70 | 48.71 | 1.27 |
| ControlNet + SPADE | 97.51 | 97.25 | 97.65 | **97.72** | 0.21 | 55.10 | 55.14 | 58.98 | 58.29 | 2.05 |

Table 2: **Results on VISOR_cond and Object Accuracy, split across the 4 spatial relation types**. $\sigma_{Vc}$ and $\sigma_{OA}$ denote the respective metric's standard deviation w.r.t all spatial relations, per method. We find that regardless of the spatial relation, SPADE enables T2I models to consistently produce spatially accurate images, a challenge faced by earlier approaches.

## 5 EXPERIMENTAL RESULTS

### 5.1 VISOR METRIC AND DATASET

Given an input prompt containing 2 objects and a spatial relationship between them, the VISOR metric evaluates the accuracy of the generated image. Object Accuracy (OA) calculates if both objects are present in the generated image. Conditional Visor (Visor$_{cond}$) quantifies the probability of relationship correctness, given both objects were correctly generated whereas Unconditional Visor (Visor$_{uncond}$) measures if the model can generate both objects and maintain the spatial relationship. VISOR$_n$ is the probability that at least $n$ out of $N$ images will have VISOR=1 for a given text prompt. The VISOR dataset contains 25,280 sentences describing two-dimensional spatial relationships. For each sentence in VISOR, we sample a corresponding image from our SPADE database.

### 5.2 EXPERIMENTAL SETUP

We perform experiments on 3 variants of Stable Diffusion (SD), versions 1.4 [4], 1.5 [5] and 2.1 [6]. We use the canny edge conditioned checkpoint [7] for ControlNet experiments.

The reference and generated RGB images are of dimension $[512, 512]$. The number of denoising steps for Stable Diffusion are varied in the range $[20 - 35]$ , while it is fixed at 20 for ControlNet.

---

[4]`https://huggingface.co/CompVis/stable-diffusion-v1-4`

[5]`https://huggingface.co/runwayml/stable-diffusion-v1-5`

[6]`https://huggingface.co/stabilityai/stable-diffusion-2-1`

[7]`https://huggingface.co/lllyasviel/sd-controlnet-canny`

| Model | Background | IS (↑) | OA (%) | VISOR (%) | | | | | |
|---|---|---|---|---|---|---|---|---|---|
| | | | | uncond | cond | 1 | 2 | 3 | 4 |
| SD 1.4 | White | 16.16 | 53.96 | 52.71 | 97.69 | 77.79 | 61.02 | 44.9 | 27.15 |
| | Indoor | 19.11 | 48.53 | 45.12 | 92.97 | 74.82 | 53.79 | 34.78 | 17.09 |
| | Outdoor | **20.16** | 44.32 | 41.80 | 94.31 | 69.79 | 49.38 | 31.86 | 16.17 |
| SD 1.5 | White | 16.27 | **54.33** | **53.08** | 97.72 | **78.07** | **61.27** | **45.44** | **27.55** |
| | Indoor | 19.11 | 48.77 | 45.28 | 92.85 | 74.93 | 53.96 | 34.77 | 17.47 |
| | Outdoor | 19.66 | 43.99 | 41.51 | 94.36 | 69.48 | 48.58 | 31.46 | 16.52 |
| SD 2.1 | White | 12.79 | 48.26 | 47.11 | 97.61 | 76.07 | 55.75 | 37.10 | 19.53 |
| | Indoor | 11.52 | 31.08 | 29.37 | 94.50 | 59.80 | 33.96 | 17.40 | 6.34 |
| | Outdoor | 10.51 | 36.37 | 34.67 | 95.34 | 65.05 | 41.23 | 23.05 | 9.36 |

Table 3: **The quantitative impact of backgrounds in SPADE images on LSAI methods**. We discover the overall best performance on VISOR is obtained when using the white background, while using the outdoor background yields the most diverse outputs in term of the Inception Score.

The baselines we compare against are Stable Diffusion (SD 1.4), GLIDE, DALLE-Mini (Dayma et al., 2021), DALLE-v2 (Ramesh et al., 2022), Control GPT and Layout Guidance. For holistic evaluation, we also report the Inception Score (IS) (Salimans et al., 2016), wherever applicable. For all subsequent tables, **Bold** values denote best performance while underlined values indicate the second-best performance.

## 5.3 MAIN RESULTS

We summarize our representative results in Table 1 and Table 2. Results are shown considering images from SPADE with a white background and # of denoising steps = 30 (for SD). Compared against existing open-source models, we achieve a $\Delta$ improvement of 17.69% and 25% in OA and Visor$_{uncond}$, respectively. The high Visor$_{cond}$ value denotes that whenever we are able to generate objects correctly, they are majorly in the right spatial orientation. More interestingly, through SPADE, we are able to increase the likelihood consistency of generating spatially correct images, as can be seen the relatively high value of Visor$_4$. Table 2 depicts that unlike other methods, we are able to maintain a high and constant Visor$_{cond}$ score, irrespective of the spatial direction. For example,

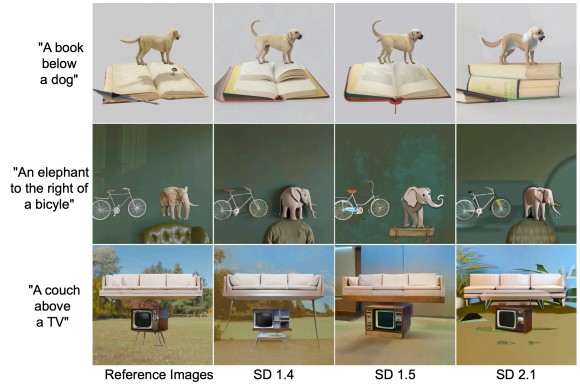

Figure 5: Illustrative example depicting the variation of generated images across the three variants of backgrounds in SPADE. A positive correlation can be seen between image diversity and background-level complexity of the initial reference image.

the largest deviation in Visor$_{cond}$ performance for ControlNet + SPADE is 0.21% between below and left relationships; in comparison Control-GPT deviates as much as 6.8% for the same.

## 5.4 IMPACT OF BACKGROUND

In Table 3, we enumerate the impact of backgrounds in the SPADE Images and the consequent trade-off between VISOR performance and model diversity. Utilizing white backgrounds that exclusively feature the two objects in question minimizes potential distractions for the model. Conversely, when the model is presented with SPADE images incorporating indoor or outdoor backgrounds, it exhibits the capacity to identify and leverage *distractor* objects, resulting in the generation of diverse images. As depicted in Figure 5, it is evident that, while all the generated images maintain spatial accuracy,

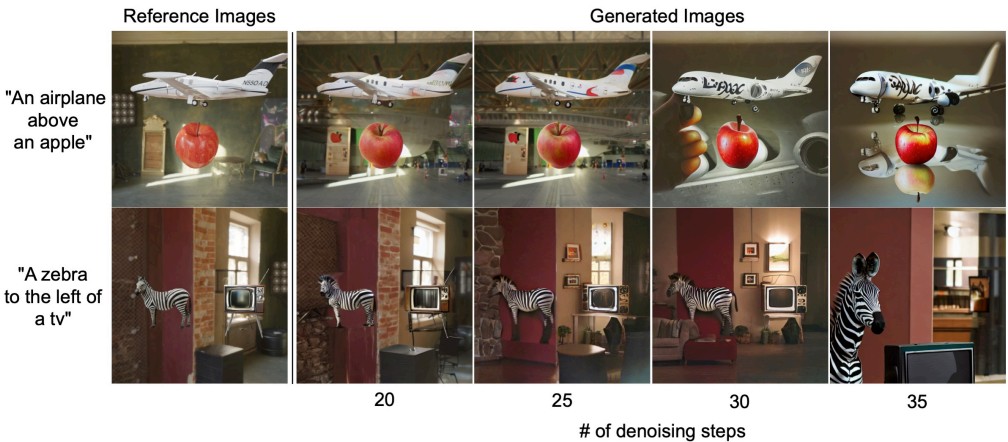

Figure 6: As additional noise is added, the generated images noticeably diverge from the reference image. Even with increased noise, our method consistently demonstrates the ability to accurately position objects. We experiment on SD 1.4 with an indoor background for these results.

reference images with with a higher degree of noise yield a greater degree of distinctiveness in the generated images.

## 5.5 CONTROLLABILITY VS PHOTO-REALISM

In this setup, we rigorously study the impact of the # of denoising steps as described in Section 4. We expectantly find that as more noise is added, the performance on VISOR deteriorates along with the deviation from the reference image. As shown in Figure 7, we find an inverse relationship between the model's ability to be diverse and maintain spatial relationship. Illustratively in Figure 6 we find that while the reference image and the generated images progressively diverge, they are able to maintain spatial relationship throughout.

Through attention activation patterns (Figure 8) during the denoising process, we find that SPADE enables better localization, at an object and spatial level. Due to space limitations, we provide ControlNet, GPT-4 and object-level results in Appendix A

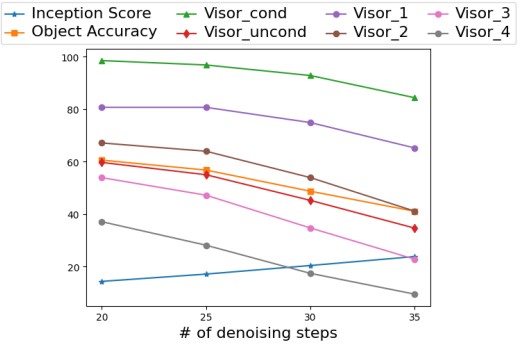

Figure 7: With an increase in the number of denoising steps, controllabillity of the models decrease with an increase in the Inception Score ; through our method we are able to improve the trade-off and attain better spatial fidelity.

## 5.6 HUMAN STUDIES

To understand the generalization of our method, we conduct 2 distinct experiments and perform human evaluation. We randomly sample 200 generated examples for each setup and average the scores across 4 workers. We also report unanimous (100%) and majority (75%) agreement between workers for each setup :

**Multi-Object and Directional Prompts -** In this setup, our prompt consists of 2 sentences and a corresponding reference image, covering 3 objects and 2 distinct relationships. For every image, we ask evaluators to rate if one or both sentences are correctly represented. We achieve an accuracy of 79.62% when at-least 1 sentence is correct and an accuracy of 46.5% when the entire prompt is correctly represented. The unanimous and majority agreement between the workers was found to be 64.5% and 86.5% respectively. While these results are comparable to other models' ability to follow

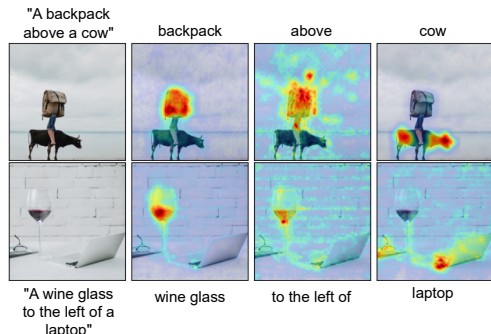

Figure 8: Illustration of accurate attention activation maps corresponding to a generated Image. SPADE provides better object localization and spatial guidance during the diffusion denoising step.

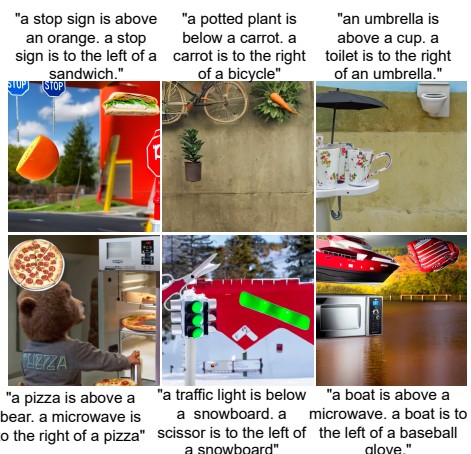

Figure 9: Illustrative example of leveraging SPADE to generate spatially correct images with 3 objects and 2 relations.

1 sentence, we emphasize scope of improvement in this regard. Illustrative examples are presented in Figure 9.

**Out-of-Distribution Objects -** In this scenario, we consider prompts containing exactly one object *not found* in SPADE. For the reference image, we locate the corresponding MS-COCO object w.r.t the OOD object by the list in Appendix D and create a reference image with the in-distribution substitute. As is shown in Figure 10, we find that our method is sufficiently able to generate a spatially faithful image using a textual prompt with an OOD object along with the reference image that has the corresponding substitute, e.g. generating for "a helicopter above a bicycle" by giving a reference image of "an airplane above a bicycle". For human evaluation, we ask workers to rate 0/1 for wrongness/correctness respectively. We attain an accuracy of 63.62% with a unanimous and majority agreement of 67% and 90.5% respectively.

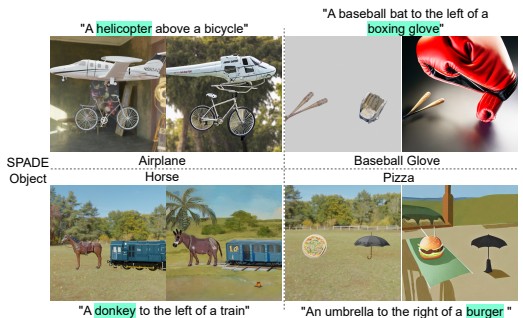

Figure 10: We find that our method is able to correctly generate and position objects not in our database. OOD Objects are highlighted. For each pair, left is the reference image and right is the generated image.

## 6 CONCLUSION

In this work we introduce SPADE, a large-scale database to improve the spatial fidelity of Text-to-Image generative models. We find that by leveraging SPADE, we can achieve state-of-the art performance on standard benchmarks as well as obtain better generalization and robustness in comparison to other methods. Most importantly, our approach is fully automated, inexpensive, and requires no manual intervention. SPADE can also serve multiple purposes: it can function as a probing dataset for assessing the spatial reasoning capabilities (Liu et al., 2023b) of multimodal large language models (Appendix F.1), and it can also serve as a valuable resource for data augmentation in the context of contrastive learning (Purushwalkam & Gupta, 2020). We also envision expanding SPADE to serve as a versatile framework capable of generating reference images for a wide range of computer vision tasks. Finally, we hope that our method is another step in the right direction towards development of safer and intelligent generative models.

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

# A APPENDIX - ADDITIONAL RESULTS

## A.1 RESULTS ON CONTROLNET + SPADE

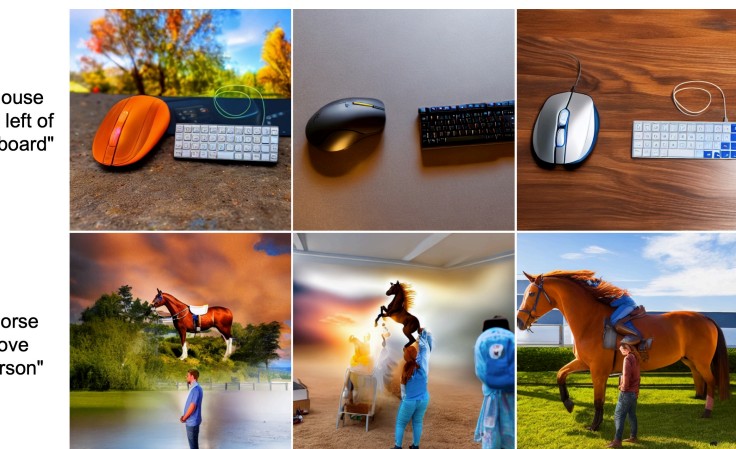

"A mouse to the left of a keyboard"

"A horse above a person"

Figure 11: **Illustrative examples generated with ControlNet + SPADE**. In real life, we are more likely to find cases of *"a person above a horse"*. However, through our method, we are able to generate such rare-case images that look genuinely convincing.

| Background | IS | OA (%) | VISOR (%) | | | | | |
|---|---|---|---|---|---|---|---|---|
| | | | uncond | cond | 1 | 2 | 3 | 4 |
| White | **18.82** | 56.88 | 55.48 | 97.54 | 78.82 | 62.93 | 48.58 | 31.59 |
| Indoor | 14.75 | **59.64** | **58.08** | 97.39 | **81.38** | **66.35** | **51.27** | 33.33 |
| Outdoor | 16.20 | 56.54 | 56.22 | **99.45** | 75.97 | 62.99 | 50.57 | **35.37** |

Table 4: **ControlNet + SPADE results on VISOR**.

The ControlNet-based results are presented in Table 4. We find that, a) we achieve the best trade-off between IS and VISOR for ControlNet and b) compared to SD, we are able to achieve higher $Visor_4$ score, indicating correctness over multiple trials. Moreover, we are able to quantify that our SPADE images have enough low-level information to faithfully represent spatial orientations and not contain any biases from our original assets, as can be seen in Figure 11.

## A.2 GPT-4 GUIDED GENERATION

Table 5 shows results of performing conditioning using GPT-4 as the alternative Coordinate Generator. Compared to our baseline results in Table 1, we notice an average of 10-point drop in performance in both Object Accuracy and $VISOR_{uncond}$ scores. These patterns develop as a result of GPT-4's propensity to generate reference images where the two objects collide into each other and become indistinguishable. Hence, T2I models tend to ignore either object, which correspondingly leads to lower VISOR scores.

| Method | OA | VISOR | |
|---|---|---|---|
| | | uncond | cond |
| SD 1.4 + SPADE | 43.88 | 41.18 | 93.85 |
| SD 1.5 + SPADE | **44.35** | **41.64** | 93.89 |
| SD 2.1 + SPADE | 39.03 | 36.86 | **94.43** |

Table 5: **VISOR Results on using GPT-4 as the Coordinate Generator**. The drop in performance is attributed to the proclivity of GPT-4 to place both the objects too close to each other in the coordinate space.

### A.3 OBJECT-WISE SPATIAL ACCURACY ANALYSIS

In Figure 14, we report which objects in MS-COCO are more likely to be correctly generated via our SPADE-based T2I method. On average, we now have a 61% likelihood that an MS-COCO object can be faithfully positioned in the output image.

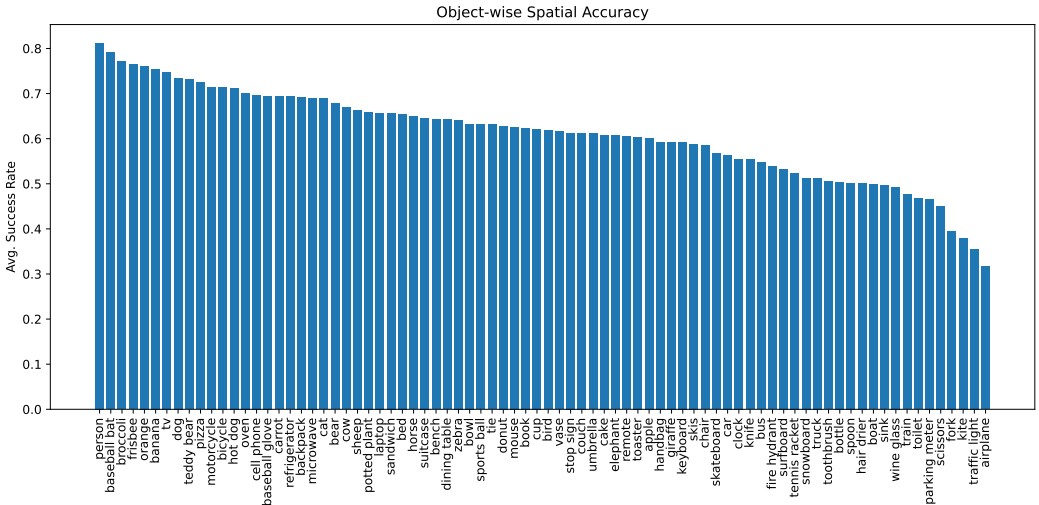

Figure 12: Average Success Rate of each SPADE object being spatially correct according to the input prompt in the generated image. We report results using the white background with SD v1.5.

### A.4 ADDITIONAL ILLUSTRATIONS

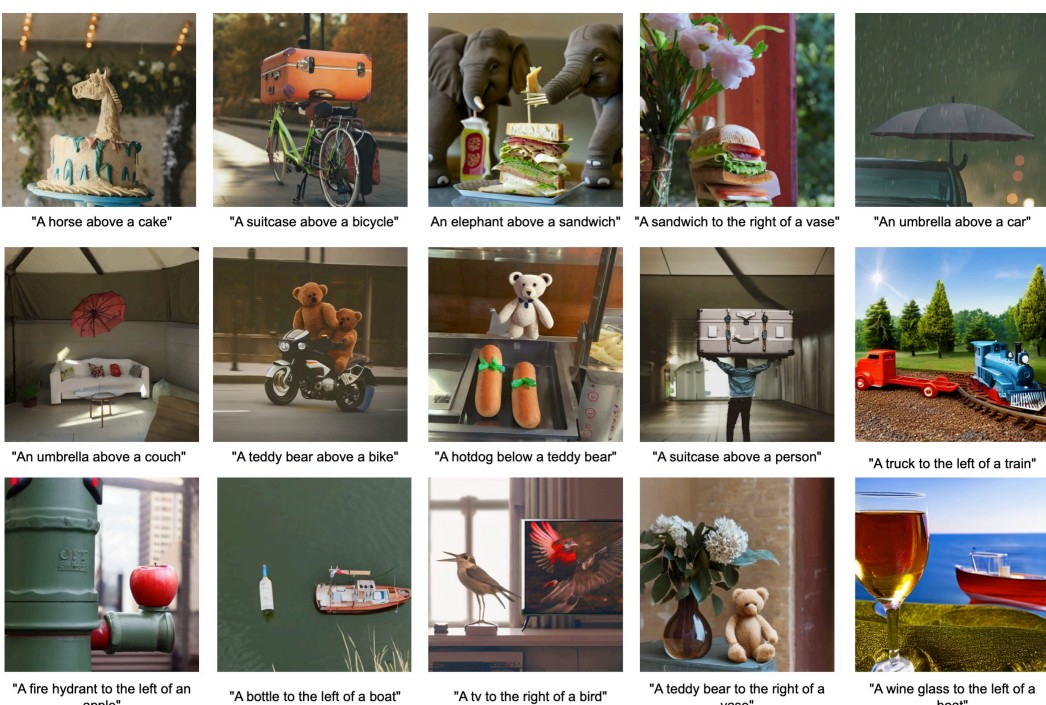

Figure 13: Correctly generated Images from T2I models by leveraging SPADE as additional guidance.

## A.5 FAILURE CASES

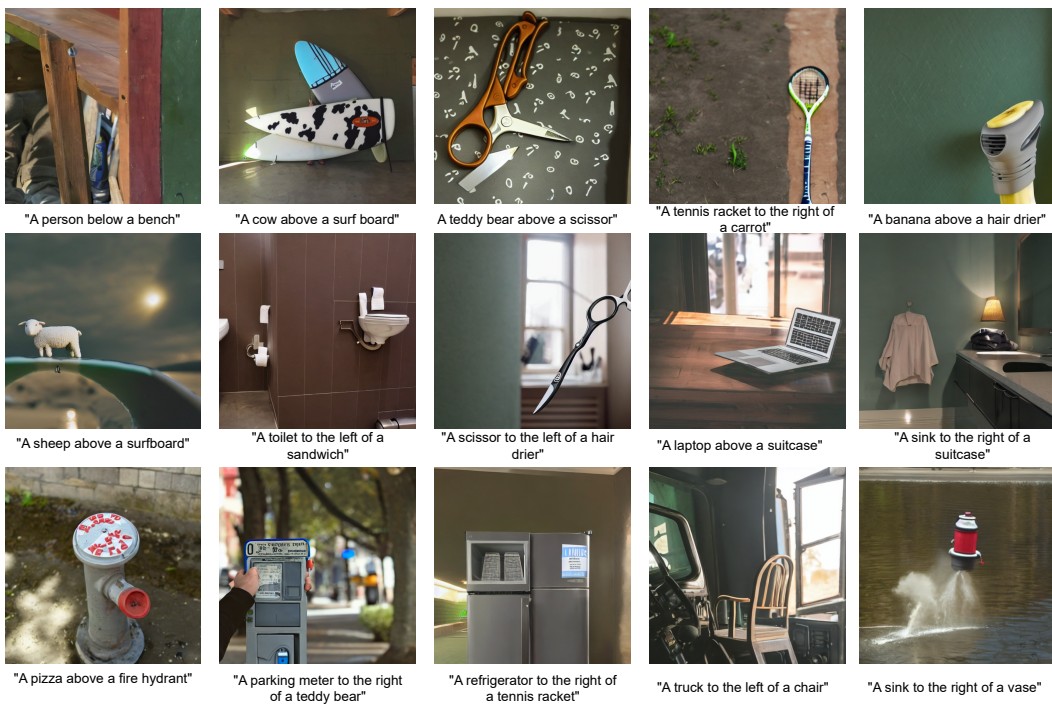

Figure 14: Images generated from T2I models by leveraging SPADE, which either do not have correct objects or are spatially incorrect

# B  APPENDIX - SPADE DATASET SHOWCASES

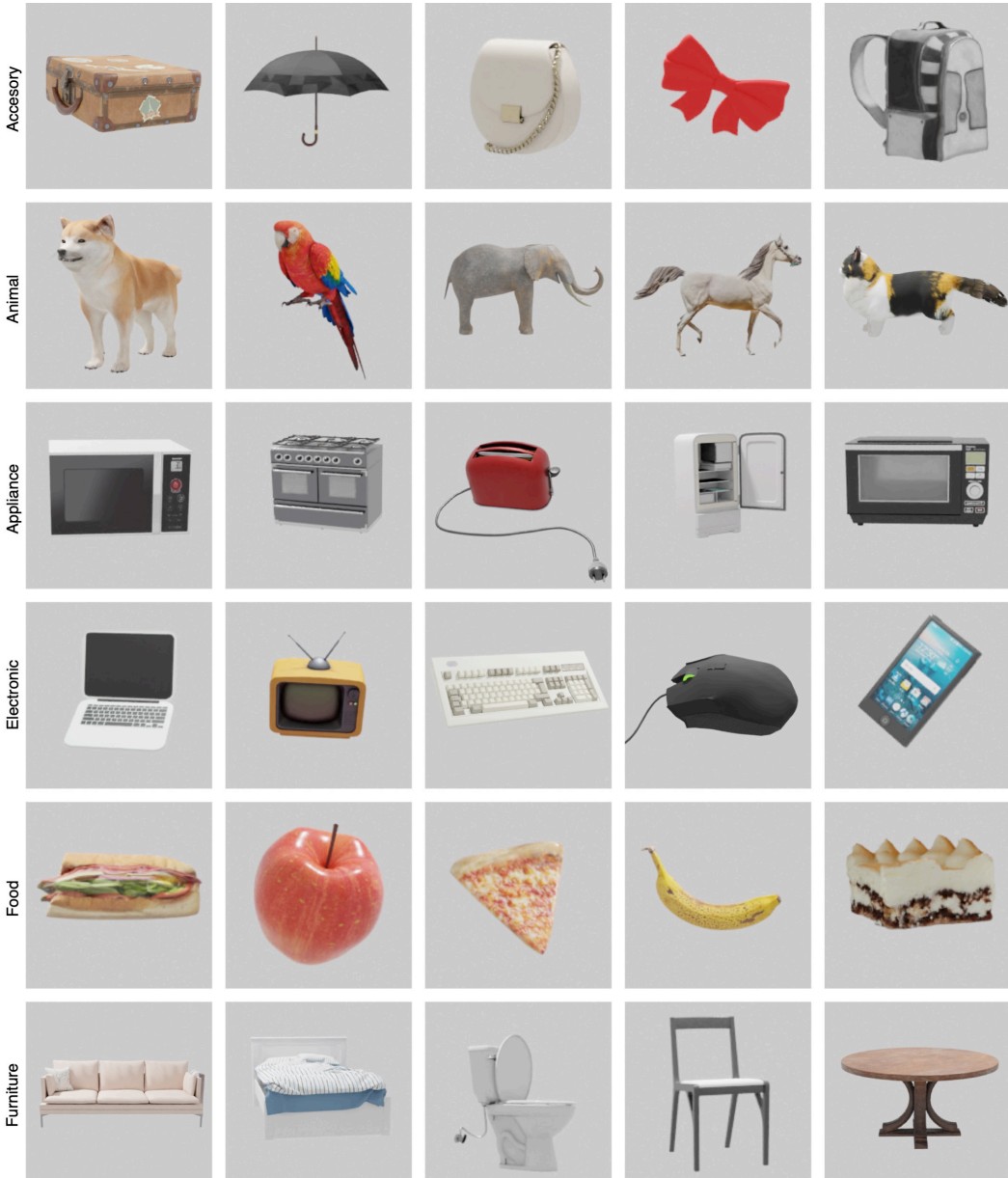

Figure 15: Example 3D object assets used in SPADE in categories of MSCOCO.

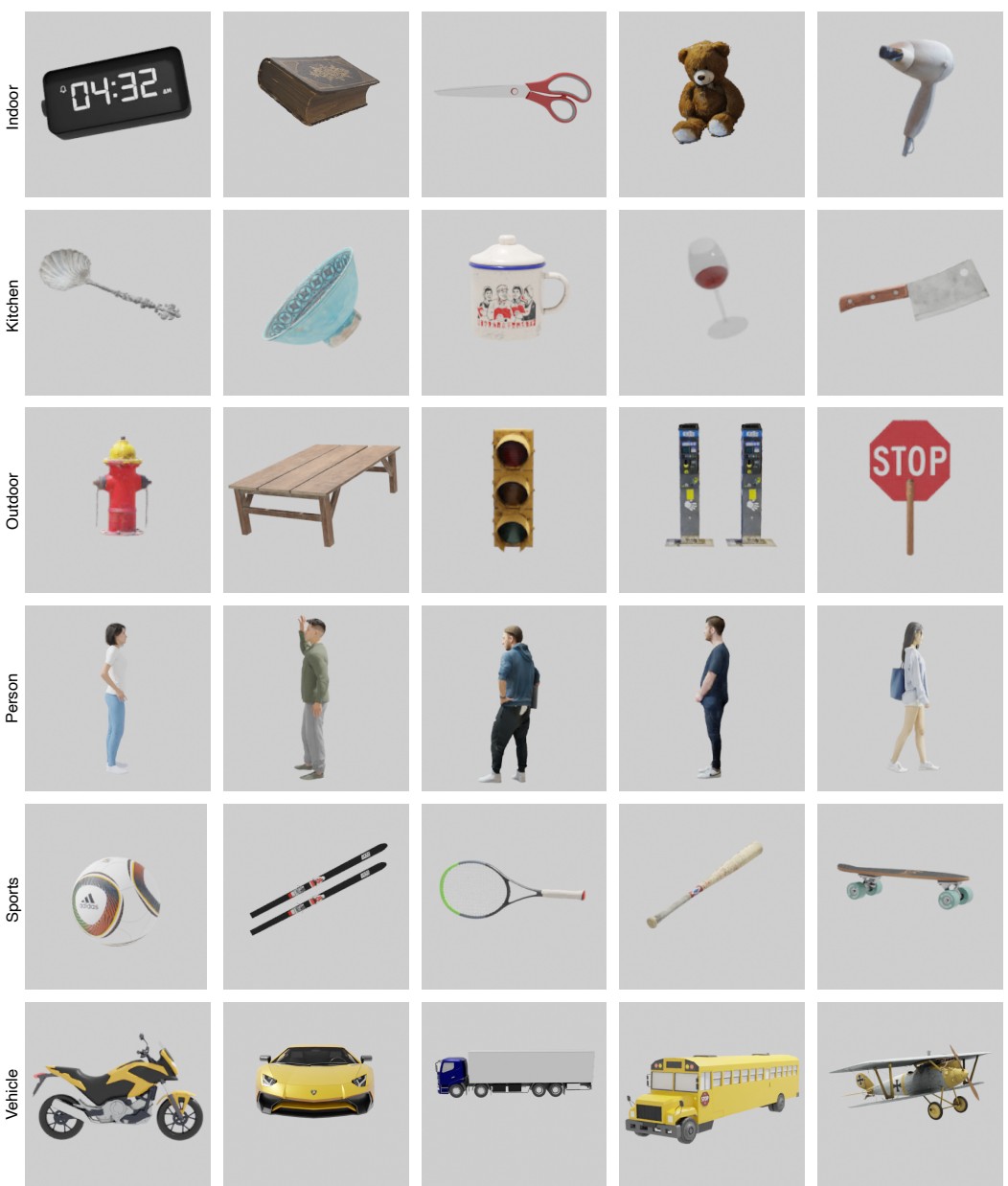

Figure 16: Example 3D object assets used in SPADE in categories of MSCOCO, continued.

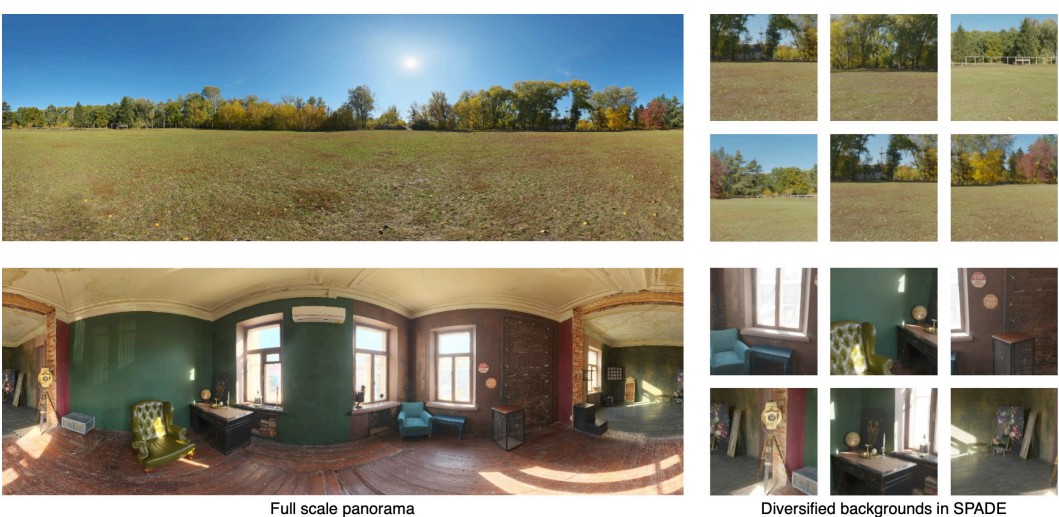

Full scale panorama                    Diversified backgrounds in SPADE

Figure 17: The 2 non-solid-white 360-degree panorama images used for square backgrounds in SPADE - Outdoor (above) and Indoor (below).

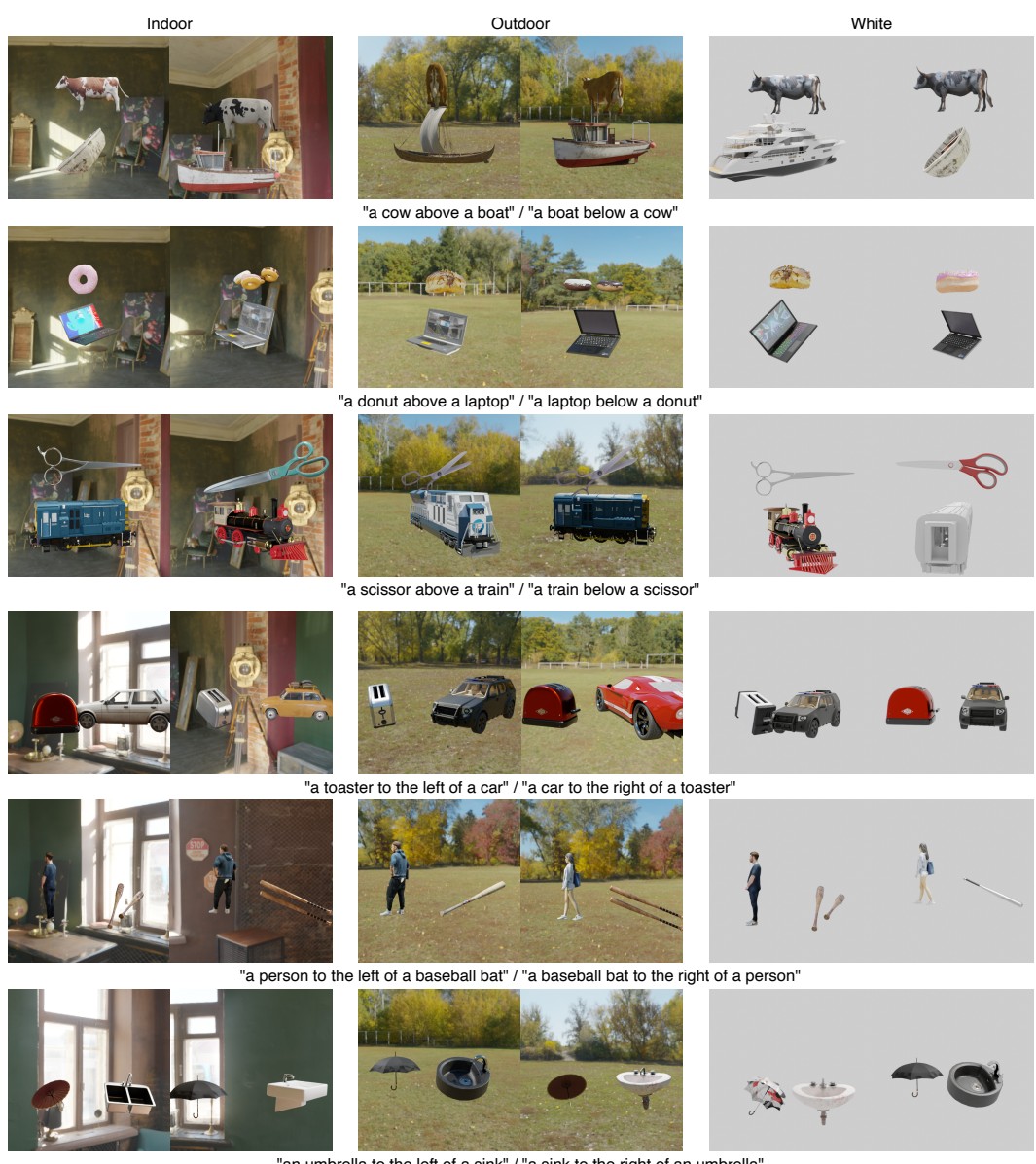

Figure 18: Additional examples of synthesized reference images in SPADE varying in spatial relations, backgrounds, and object assets.

## C  APPENDIX - GPT-4 PROMPTS FOR REFERENCE IMAGE SYNTHESIS

When using GPT-4 as the Coordinate Generator module in the SPADE Generator, we use the System Prompt in Figure 19 as the global context. It is paired with each individual Input Prompt in Table 6 to generate two sets of coordinates for the two objects mentioned in the Input Prompt.

> *"I will give you certain sentences, and you will have to generate (Y, Z) coordinates of the centroid of the objects given in the sentence. Y coordinate determines left-right and Z coordinate determines above-below.*
>
> *Rules :*
> *1. Negative Y means left, Positive Y means right.*
> *2. Negative Z means bottom, Positive Z means above.*
> *3. All coordinate values must be between (-100, 100).*
>
> *For example,*
> *1. An Apple is above a Person = ((3,80),(-10,5))*
> *2. A train is below a cat = ((9, -84), (17, -10))*
> *3. An elephant is to the left of an airplane = ((-97, 1), (89, -7))*
> *4. A bottle is to the right of a bike = ((87, -9), (-18, 1))*
>
> *Be as varied as possible, and keep in mind the relative sizes of the mentioned objects in the sentence. Now, generate output for the following sentence."*

Figure 19: The global contextual System Prompt for generating coordinates with input text that describes a scene of 2 objects and 1 spatial relation. We only concern object coordinates on Y and Z axes, as we view the objects from a point along the positive side of X axis in the SPADE Generator.

| Input Prompt | GPT-4 Generated Coordinates | | | |
|---|---|---|---|---|
| | Object 1 | | Object 2 | |
| a cow is above a cup | cow | (5, 79) | cup | (-6, 0) |
| a broccoli is above a bus | broccoli | (15, 90) | bus | (-50, 10) |
| a knife is below an airplane | knife | (-5, 2) | airplane | (14, 86) |
| a couch is below an apple | couch | (-15, 2) | apple | (6, 90) |
| a dining table is to the left of a skateboard | dining table | (-87, 5) | skateboard | (20, -2) |
| an elephant is to the left of a hair drier | elephant | (-85, 2) | hair drier | (30, 95) |
| a vase is to the right of a fork | vase | (10, 20) | fork | (-70, 3) |
| a potted plant is to the right of a person | potted plant | (25, 30) | person | (-75, 10) |

Table 6: Example Input Prompts with their respective object coordinates generated by GPT-4.

## D   APPENDIX - OUT-OF-DISTRIBUTION OBJECT SUBSTITUTES

For experiments involving out-of-distribution objects, we swap the one OOD object in the input prompt with its corresponding MS-COCO substitute object in Table 7 when creating reference images using the SPADE Generator.

| MS-COCO Object | OOD Object | MS-COCO Object | OOD Object |
| --- | --- | --- | --- |
| airplane | helicopter | kite | flag |
| apple | pear | knife | sword |
| backpack | purse | laptop | tablet |
| banana | mango | microwave | toaster oven |
| baseball bat | walking stick | motorcycle | tractor |
| baseball glove | boxing glove | mouse | webcam |
| bear | monkey | orange | papaya |
| bed | table | oven | dishwasher |
| bench | sofa | parking meter | phone booth |
| bicycle | scooter | person | mannequin |
| bird | butterfly | pizza | burger |
| boat | submarine | potted plant | tree |
| book | magazine | refrigerator | cabinet |
| bottle | lunchbox | remote | game controller |
| bowl | plate | sandwich | salad |
| broccoli | cauliflower | scissors | pliers |
| bus | tram | sheep | goat |
| cake | pie | sink | bathtub |
| car | ambulance | skateboard | roller skates |
| carrot | sweet potato | skis | hockey stick |
| cat | rabbit | snowboard | sled |
| cell phone | landline phone | spoon | straw |
| chair | barstool | sports ball | bowling ball |
| clock | wall calendar | stop sign | parking sign |
| couch | cushion | suitcase | duffel bag |
| cow | panda | surfboard | kayak |
| cup | tumbler | teddy bear | doll |
| dining table | dressing table | tennis racket | badminton racket |
| dog | fox | tie | bowtie |
| donut | pudding | toaster | waffle maker |
| elephant | lion | toilet | shower |
| fire hydrant | mailbox | toothbrush | comb |
| fork | chopsticks | traffic light | streetlight |
| frisbee | basketball | train | roller coaster |
| giraffe | camel | truck | crane |
| hair drier | hairbrush | tv | computer monitor |
| handbag | cardboard box | umbrella | tent |
| horse | donkey | vase | pitcher |
| hot dog | burrito | wine glass | glass jar |
| keyboard | piano | zebra | llama |

Table 7: Substitute OOD object nouns for the original 80 MS-COCO objects used in SPADE.

## E   APPENDIX - CHOICE OF DENOISING STEPS

We choose the denoising steps in the range $[20, 35]$ as suggested by the authors in the original SDEDit (Meng et al., 2022) work. The authors empirically find that setting $t_0 \in [0.3, 0.6]$, (which translates to the above range) leads to optimal results across a wide range of tasks.

## F APPENDIX - POTENTIAL FUTURE USAGE OF SPADE

### F.1 EVALUATION OF MULTIMODAL LLMS

We may explore using SPADE as a benchmark to evaluate Multimodal Large Language Models (MLLM) on the capability of spatial fidelity. Table 8 demonstrates such a use case where we use SPADE for the task of Visual Question Answering (VQA). Our preliminary results indicate that the current state-of-the-art MLLMs may still struggle to extract correct spatial relations given a multi-object synthetic scene image.

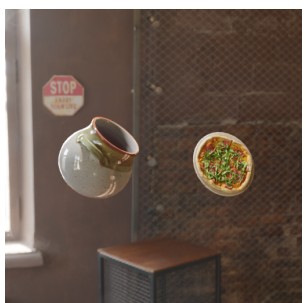 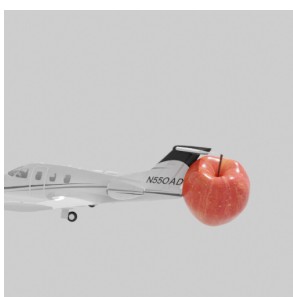 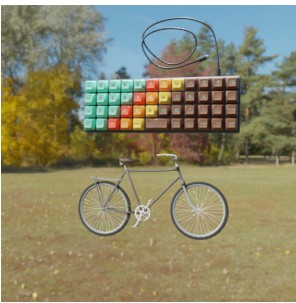

| | | | |
|---|---|---|---|
| Question: | Does the image show a pizza on the right of a vase? | What is the object on the left? | Is the keyboard below the bicycle? |
| Answer: | Yes, the image shows a pizza on the right of a vase. ✓ | The object on the left is a small airplane. ✓ | Yes, the keyboard is below the bicycle. ✗ |
| Question: | Does the image show a vase on the left of a pizza? | What is the object on the right? | Is the bicycle above the keyboard? |
| Answer: | Yes, the image shows a vase on the left of a pizza. ✓ | The object on the right is an apple. ✓ | Yes, the bicycle is above the keyboard, as it is suspended in the air. ✗ |

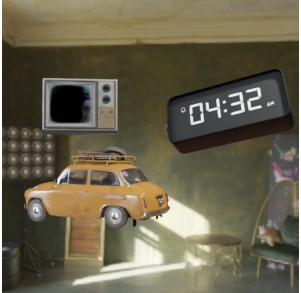 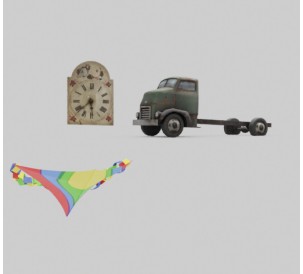 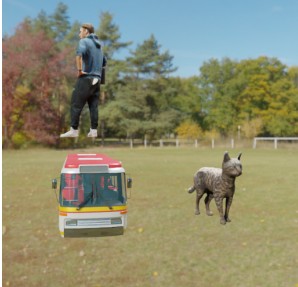

| | | | |
|---|---|---|---|
| Question: | What is above the car in the image? | Does the image depict a truck on the left of a clock? | Is there a cat to the right of a bus? |
| Answer: | Above the car in the image, there is a TV and a clock. ✓ | Yes, the image depicts a truck on the left of a clock. ✗ | Yes, there is a cat to the right of a bus. ✓ |
| Question: | What is on the right of the tv in the image? | Does the image depict a truck on the right of a clock? | Is there a cat above a bus? |
| Answer: | There is a car on the right of the tv in the image. ✗ | Yes, the image depicts a truck on the right of a clock. ✓ | Yes, there is a cat above a bus in the image. ✗ |

Table 8: Example VQA performance using images created by the SPADE Generator. The textual answers are generated by LLaVA (Liu et al., 2023a). ✓ indicates the generated answer matches with its input's ground truth, while ✗ indicates otherwise.

### F.2 CONTRASTIVE LEARNING

SPADE pairs multiple reference image variants with one textual description (input prompt). Taking advantage of such feature, we may also explore leveraging SPADE for multi-modal tasks that involve self-supervising contrasive learning. In line with fundamental works such as Siamese Network (Koch et al., 2015) and SimCLR (Chen et al., 2020b), we are now able to collect positive samples and negative samples w.r.t an anchor image from the SPADE dataset with ease. We would like to see

if SPADE helps achieve more robust visual representation extraction by optimizing the constractive loss (Hermans et al., 2017), thus potentially leading to better performances on downstream tasks such as object detection, image classification, or more.

## G    APPENDIX - LIMITATIONS

Although we made every effort to maintain as much diversity as possible while developing SPADE and gathering the object assets, there may still be misses in this regard. Our current scope consists of a finite number of MS-COCO objects, backgrounds and 2D relationships; however we believe that all of the above aspects are easily extendable. Since, our method leverages pre-trained T2I models, we also inherit their shortcomings in terms of biases and instability.

