# OpenReview forum: "Spade : Training-Free Improvement of Spatial Fidelity in Text-to-Image Generation"
_ICLR.cc/2024/Conference — ICLR 2024 Conference Withdrawn Submission_

### Official Review · Reviewer_2sKk · 2023-10-28

**Soundness:** 2 fair
**Presentation:** 2 fair
**Contribution:** 2 fair
**Rating:** 3
**Confidence:** 4

**Summary:**

This work aims at generating accurate spatial relations between objects with a text-to-image generative model. The work first introduces SPADE, a large database of synthetic text-image pairs from open-source 3D renderings. The work then uses LSAI, a training-free text-to-image generator that leverages SPADE to provide additional guidance for text-to-image generative models in order to achieve high spatial fidelity. With SPADE and LSAI, the method is able to generate spatially faithful images and outperforms previous works on the VISOR benchmark.

**Strengths:**

1. This work proposes a pipeline to artificially synthesize databases that have a specific spatial relationship. The resulting database, SPADE, can then be leveraged as a conditioning for downstream generation.
2. This work proposes a conditional generator that takes in SPADE generation as a reference for SDEdit and ControlNet in order to generate images to have layouts according to the prompt specification.
3. This work outperforms previous approaches in terms of the accuracy of following the input prompts that include spatial keywords on text-to-image benchmark VISOR.

**Weaknesses:**

1. The authors listed ControlGPT [1] and LayoutGuidance [2] in the introduction section to demonstrate the point that existing approaches either require expensive training or labeled annotations. However, this summarization seems to miss two related works: LLM-grounded Diffusion [3] and LayoutGPT [4]. While the pipeline of [4] requires a reference annotated dataset, the pipeline of [3] does not include training (other than stable diffusion training itself) or using extra labeled annotations. Furthermore, this work uses a pre-collected 3D asset library, which adds an extra requirement that previous methods [1,2,3,4] do not need. Since the requirements of [3] are similar to this work (i.e., Stable Diffusion and GPT-4 without additional training), an additional discussion between the current work and [3] is needed. The authors are also encouraged to compare with [3,4] in terms of spatial fidelity.
2. The work proposes a database named SPADE as one of the main contributions, but the database has 80 MS-COCO classes, with hand-collected 3D model assets and limited 2D spatial relationships. Therefore, it is hard to know whether the proposed LSAI method, based on SPADE, can generalize to categories that are rare and relationships that are different from the ones in the dataset (e.g., A is far away from the camera than B).
3. VISOR only considers simple relationships (e.g., above/under/left/right) with limited classes (COCO classes), which the synthetic database considers. More evaluation on challenging benchmarks that test diverse object types and relationships, such as the spatial part of [5], is needed to explain the improvements of this method in more challenging scenarios.
4. The user study only asks the users questions about whether the generation is correct. However, whether this method will introduce degradation in terms of visual quality is also a very important part in the evaluation. If this is not considered, then an example with simple copy-and-paste can achieve high scores in the evaluation.
5. This work lacks clarity in presentation in the method introduction. For example, it is unclear how Eq. 1 relates to LSAI in addition to being a part of the standard denoising diffusion framework. It is also unclear how the proposed LSAI, which leverages SPADE generator that requires explicit spatial keywords and object names, is able to parse arbitrary user inputs, including those that do not have spatial keywords or have other types of special keywords (e.g., near/far/inside/outside).
6. This work seems to generate floating objects that do not seem to compose well. For example, in Fig. 5, shadows are missing in the generated image.

[1] Controllable Text-to-Image Generation with GPT-4. https://arxiv.org/abs/2305.18583
[2] Training-free layout control with cross-attention guidance. https://arxiv.org/abs/2304.03373
[3] LLM-grounded Diffusion: Enhancing Prompt Understanding of Text-to-Image Diffusion Models with Large Language Models. https://arxiv.org/abs/2305.13655
[4] LayoutGPT: Compositional Visual Planning and Generation with Large Language Models. https://arxiv.org/abs/2305.15393
[5] T2I-CompBench: A Comprehensive Benchmark for Open-world Compositional Text-to-image Generation. https://arxiv.org/abs/2307.06350

**Questions:**

* Are these 3D assets hand-crafted or captured? The authors are encouraged to explain whether the effort in designing or capturing the 3D assets is lower or comparable to annotating images that are used by previous methods such as [4] to generate images.
* Could LSAI generate objects that do not have a corresponding mapping to objects in COCO (e.g., very different from these categories)?

---

### Official Review · Reviewer_dk9K · 2023-10-31

**Soundness:** 2 fair
**Presentation:** 2 fair
**Contribution:** 1 poor
**Rating:** 5
**Confidence:** 4

**Summary:**

This paper presents an approach for text2image synthesis using pre-trained diffusion model with an emphasis on spatial fidelity (i.e. respecting spatial arrangement of objects specified in the input text). To that end, a database of synthetic images is first generated. Next, synthetic images are used as guidance during the diffusion process (SDEdit for guide stable diffusion, and edge map to guide ControlNet).

**Strengths:**

- The idea is straightforward to understand, and the generation pipeline is simple.
- The paper is well-structured and well-written.
- Results indicate an improvement over (spatially unconstrained) baselines.

**Weaknesses:**

- The technical contribution is limited. ControlNet as well as SPEdit are used on top of a synthesized scene using Blender.
- The method is limited to 2 objects and 1 spatial relation.
- Missing comparison with (cited) other layout-guided diffusion models. It is obvious that the presented model will achieve better scores when evaluated on spatial fidelity against purely text2image models.
- The generated images are similar to the reference images and hence often look unrealistic and stitched (unrealistic composition of objects and background).

Side note: SPADE is not a good name, the first thing that comes to mind is the very influential semantic image synthesis model https://arxiv.org/abs/1903.07291

**Questions:**

-

---

### Official Review · Reviewer_xFgb · 2023-10-31

**Soundness:** 3 good
**Presentation:** 3 good
**Contribution:** 2 fair
**Rating:** 3
**Confidence:** 3

**Summary:**

This paper provides a method for generating spatial relationship-aware text-to-image generative models in a training-free style. The paper observes that text-to-image generative models suffer from long texts that involve multiple objects and objects with complicated spatial relationships. To resolve this problem, this paper proposes a text-image reference database called SPADE, and an image generation method(LSAI) which leverages a spatial-aware image database. Specifically, SPADE is a database with four types of 2D relationships (above, under, right, left) and 80 objects. It uses a coordinate generator and a position diversifier to generate diverse 2D spatial relationships, and a scene synthesizer to synthesize multiple backgrounds and render them into photorealistic images. The dataset contains a total of 189,600 text-image pairs. The LSAI generation process takes a text prompt as input and retrieves a reference image from the SPADE database. It then uses a guided diffusion method or a semantic-controlled diffusion model to generate a spatial-aware image from the reference image. The experiment shows the method is able to synthesize images with multiple objects and spatial positions, and is able to generalize well to OOD objects. It presents how the background and denoising steps affect the controllability of the generated images. The paper also shows that LSAI brings significant improvement for T2I generation methods on the VISOR benchmark. The paper also shows some potential applications for the SPADE database and LSAI, like proposing a new spatial-aware VQA benchmark.

**Strengths:**

This paper has the following strengths.
+ The paper has clear motivation. It aims to resolve the problem that the T2I generative model is not able to handle prompts with multiple objects and complex relationships.
+ The paper generates a spatial relation-aware database, SPADE, that is generated rigorously from rules and does not contain hallucination. This database, as noted by the authors, could bring potential applications including setting up a spatial relation-aware VQA benchmark.
+ This paper shows the proposed LSAI can improve the performance of current T2I methods on the VISOR benchmark. It is also able to generalize to multiple spatial relationships with multiple (OOD) objects to some extent.
+ The ablation studies show how the background and denoising steps affect the controllability of the generated images. This is practically helpful for studying the controllability of the guided diffusion generation.

**Weaknesses:**

Despite the strengths, the paper has the following weaknesses that limit the significance of the contribution

+ The diversity of spatial relationships. Though the paper presents the method as able to deal with multiple objects with multiple relationship terms, I think the spatial relationships that the paper uses are not diverse enough. Given that the dataset is generated with real 3D objects in 3D spatial coordinates, there are actually more spatial relationships than the four types given. Please take a look at [1] section 4, which includes 17 types of spatial relationships. Clearly, the proposed method can not reflect many other relationships including "inside/outside", etc. Other methods, LayoutGPT for example, are able to generate more diverse spatial relationships.
+ The diversity of objects. Though the paper presents that the model is able to generalize to OOD objects, I still doubt the diversity of the objects that the method can generate. Given the database only provides 80 objects and the generated process is based on a retrieval manner, I wonder if the dataset is able to generate imagined objects or if daily objects look significantly different than the dataset assets. For example, "a lamp next to a spider".
+ Photorealism. Despite the method's ability to perform well on the VISOR benchmark, it sacrifices too much in terms of photorealism.
+ Background controllability.  Layout-guided generated methods like layoutGPT, layout Guidance, and ControlGPT, are able to synthesize a reasonable background given the objects in the foreground. The proposed method seems to lose the controllability for background synthesis.

[1] 3D-VisTA: Pre-trained Transformer for 3D Vision and Text Alignment

**Questions:**

I think the paper proposes a very good database generation pipeline, but to improve the practical application of the method, I encourage the paper to -
1. Improve the diversity of spatial relationships, as I stated in the weakness section.
2. Increase the diversity of asset databases (like Objaverse-XL[1]), or use a generative method to generate single objects.
3. To actually conduct an experiment that using the proposed dataset can improve the performance of LLMs in spatial relation-aware VQA tasks.

[1] Objaverse-XL: A Universe of 10M+ 3D Objects

---

### Official Review · Reviewer_wpM5 · 2023-11-01

**Soundness:** 3 good
**Presentation:** 3 good
**Contribution:** 2 fair
**Rating:** 5
**Confidence:** 3

**Summary:**

The paper aims to address two limitations of previous text-to-image generation, which are the missing object mentioned in the prompt and the wrong spatial relationship between objects. The paper proposes the SPADE dataset and SPADE generator to generate reference images that align with the input prompt regarding object and spatial relationships. The paper proposes LSAI to use Stable Diffuison or ControNet with the reference image in a training-free manner to generate the final image. The results show that the proposed method can improve the spatial fidelity of Stable Diffusion and also have better generalization ability.

**Strengths:**

1. The manuscript successfully identifies and addresses two critical shortcomings in the field of text-to-image generation: the neglect of specified objects and errors in spatial arrangement.
2. A logical and effective strategy is proposed, where a reference image is generated to match the input prompt in terms of spatial relationships, and then utilized with SDEdit or ControlNet to create the final image. This approach demonstrates sound reasoning.
3. The method can improve the stable diffusion model. It also shows great performance in out-of-distribution objects.

**Weaknesses:**

1. The intuition behind SPADE is not explained well. The author should explain why the 3D rendering engine is needed here. Because the paper only focuses on vertical and horizontal direction, 2D is enough.
2. Some comparison experiment is missing. For example, LayoutGPT also focuses on the relationship issue, the comparison result should be given.
3. More visualization results are needed to validate the generalization ability. The OOD experiment is interesting, however, the author should show the result with different substitutes.

**Questions:**

Please refer to weaknesses.